# Predictive Roles of Basal Metabolic Rate and Body Water Distribution in Sarcopenia and Sarcopenic Obesity: The link to Carbohydrates

**DOI:** 10.3390/nu14193911

**Published:** 2022-09-21

**Authors:** Lizheng Guan, Tiantian Li, Xuan Wang, Kang Yu, Rong Xiao, Yuandi Xi

**Affiliations:** 1Beijing Key Laboratory of Environmental Toxicology, School of Public Health, Capital Medical University, Beijing 100069, China; 2Department of Clinical Nutrition, Peking Union Medical College Hospital, Beijing 100730, China

**Keywords:** basal metabolic rate, body water, carbohydrates, sarcopenic obesity, sarcopenia

## Abstract

Sarcopenic obesity is a new category of obesity and is a specific condition of sarcopenia. This study aimed to find the relationship of the basal metabolic rate (BMR) and body water distribution with muscle health and their prospective roles in screening for sarcopenic obesity and sarcopenia. The role of nutrients such as carbohydrates in the relationship was further detected. A total of 402 elderly subjects were recruited. Body composition was estimated by bioelectrical impedance analysis. Sarcopenia was defined by the Asian Working Group for Sarcopenia 2019. The cutoff values were determined by the receiver operating characteristic curve. Mediation analyses were performed using SPSS PROCESS. Higher BMR and BMR/body surface area (BSA) were protective factors against sarcopenic obesity (OR = 0.047, *p* = 0.004; OR = 0.035, *p* = 0.002) and sarcopenia (OR = 0.085, *p* = 0.001; OR = 0.100, *p* = 0.003) in elderly people. Low extracellular water (ECW)/intracellular water (ICW) and ECW/total body water (TBW) were negatively correlated with the skeletal muscle index (SMI). The intake of dietary carbohydrates in people with sarcopenic obesity was the lowest, but in subjects with obesity, it was the highest (*p* = 0.023). The results of the moderated mediation model showed that BMR fully mediated the positive relationship between carbohydrates and SMI, which was more obvious in the population with an abnormal body water distribution. BMR or BMR/BSA had the potential role of predicting a higher risk of sarcopenic obesity and sarcopenia. Higher BMR and lower ECW/ICW and ECW/TBW may benefit muscle health. The overconsumption of carbohydrates (especially > AMDR) might be a risk factor for obesity. Moderate dietary carbohydrate intake might promote SMI by regulating BMR and body water distribution in the elderly.

## 1. Introduction

Two of the biggest public health concerns in both developed countries and developing countries are the widespread prevalence of aging and obesity [1]. Currently, it is clear that the demographic trends in the size and proportion of elderly and obese populations are increasing at an unprecedented rate, which may be one of the major challenges to health systems in the near future [2,3].

Sarcopenia is recognized as an age-associated process that is characterized by the progressive loss of skeletal muscle mass and strength [4]. The European Working Group on Sarcopenia in Older People (EWGSOP) has updated the new definitions for different stages, including “probable sarcopenia”, “sarcopenia”, and “severe sarcopenia”. Additionally, based on data on the Asian population, sarcopenia could be identified by low muscle mass combined with low muscle strength or low physical performance, and severe sarcopenia covers reductions in all of the above according to the diagnostic algorithm of the Asian Working Group for Sarcopenia (AWGS) [4,5].

Sarcopenia often coexists with obesity, leading to a specific condition named “sarcopenic obesity” [6]. This is a new category of obesity in elderly people who have high adiposity coupled with low muscle mass [7], which means that with abnormal age-dependent muscle loss and fat accumulation, the two may act synergistically to maximize their threat to health. Importantly, sarcopenic obesity has been reported to increase the risk of metabolic impairment and physical disability more than either sarcopenia or obesity alone [8,9].

In recent years, it has been confirmed that the basal metabolic rate (BMR) is related to muscle strength and muscle-related diseases, and it depends on body composition [10,11]. Fat-free mass is an important independent variable in predicting BMR, which indicates the strong association between BMR and muscle quantity [12]. A low extracellular water (ECW)/intracellular water (ICW) ratio and a large phase angle (PA), which is determined by a low ECW/total body water (TBW) ratio, have also been proposed as indicators of high muscle quality in several studies [13,14]. It has limitations in that using BMI represents body fat distribution and the body fat percentage ratio [15]. A reduction in BMR is proven to be strongly correlated with obesity [16], and lower TBW has also been found in obese children [17]. In this case, BMR and body water distribution might be important indicators to predict and explore the potential population of sarcopenia or sarcopenic obesity.

On the other hand, nutrition is considered not only the causative factor of sarcopenia/obesity but also a preferred method for prevention and treatment. Evidence shows that healthier dietary patterns or supplementation with protein, unsaturated fatty acids, and many other active substances in foods has obvious beneficial effects on the improvement of skeletal muscle health [18]. More importantly, dietary nutrition might play a critical role not only in body fat mass but also in BMR and body water distribution [19,20]. Which dietary factors are potential risk factors for sarcopenia and sarcopenic obesity? Is there any intermediation of dietary nutrition on the relationship between sarcopenia with/without obesity and BMR or body water distribution?

The purpose of this study was to evaluate the relationship of BMR and body water distribution with muscle mass and to detect their prospective roles in screening for sarcopenia or sarcopenic obesity in a community of elderly people. Moreover, potential dietary risk factors for sarcopenia and sarcopenic obesity were explored. The association was studied to estimate whether nutrition affects sarcopenia or sarcopenic obesity through BMR and body water distribution.

## 2. Materials and Methods

### 2.1. Study Population

A total of 402 patients who attended a geriatric physical examination center from April to September in 2016 were included in the study in Hainan province.

### 2.2. Measurements

#### 2.2.1. Anthropometric Parameters

Anthropometric measurements, including the height, weight, hip circumference (HC), and waist circumference (WC) of all participants, were measured by experienced technicians. Body weight and standing height were measured using a scale accurate to 0.1 kg and 0.1 cm (Hengxing RGT-140, Jiangsu, Yixing, China). WC and HC were also measured to the nearest 0.1 cm. BMI and the waist-to-hip ratio (WHR) were then calculated.

#### 2.2.2. Body Composition

Skeletal muscle mass was estimated by bioelectrical impedance analysis (BIA) (Tanita BC-418, Tokyo, Japan). An electric current was supplied by electrodes on the tips of both toes and fingers. Subjects stepped barefoot on the weighing platform with their heels on the rear electrodes and the front part of their feet in contact with the front electrodes. They stood still without bending their knees. The subjects also grasped the grips with both hands. When the measurements were complete, the analyzer displayed the resistance. Thus, total skeletal muscle mass, fat mass, lean body mass, fat-free mass, and visceral fat area (VFA) were all measured. Total skeletal muscle mass (TSM, kg) was obtained by adding up the lean tissue mass of the arms, legs, and trunk. Appendicular skeletal muscle mass (ASM, kg) was calculated as the sum of the limb muscle mass, as described elsewhere. The skeletal muscle mass index (SMI) was obtained by dividing ASM by height squared (m^2^) [21].

BMR was calculated by BIA, which is a validated formula confirmed by indirect calorimetry measurements. In addition, we divided the BMR by height squared (cm^2^), body surface area (BSA), and BMI to exclude the effects of height and weight [11]. BSA was calculated by the formula BSA (m^2^) = square root of [height (cm) × weight (kg)/3600] [22].

#### 2.2.3. Handgrip Strength Function

Handgrip strength (HGS) data were obtained using the Jamar Plus + Digital Hand Dynamometer (Sammons Preston, Bolingbrook, IL, USA) with a resolution of 0.1 kg-force (Kgf), which was calibrated by the manufacturer [23]. The measurement was conducted in the sitting position, with the elbow flexed at 90° and the wrist in the neutral position. Participants who were in the sitting position squeezed three times as hard as possible (with a 30 s rest between each attempt), and the highest was recorded as the grip strength. Each participant performed 3 measurements with a 1 min pause between them. The maximum value of 3 consecutive measurements performed in this study was registered [24].

### 2.3. Definition of Sarcopenia and Sarcopenic Obesity

The sarcopenia stage is characterized by the combination of low muscle mass plus either low grip strength or slow gait speed, whereas severe sarcopenia is the stage when all three criteria for the definition are met (i.e., low muscle mass and both low muscle strength and low physical performance) [4]. Sarcopenia was defined as HGS < 28 kg and SMI < 7.0 kg/m^2^ for males and HGS < 18 kg and SMI < 5.7 kg/m^2^ for females by using previously reported definitions [5,25]. Obesity was defined as BMI (≥28.0 kg/m^2^) or body fat (≥25.0% for males, ≥35.0% for females) [26] or WC (≥90 cm for males, ≥80 cm for females) [27] in this study. Patients with sarcopenia and obesity were considered to have sarcopenic obesity [24,28]. Subjects were divided into four groups: non-sarcopenia/non-obesity (healthy), sarcopenia/non-obesity (sarcopenia), sarcopenia/obesity (sarcopenic obesity), and non-sarcopenia/obesity (obesity) [29].

### 2.4. Dietary Variables

Participants completed a food frequency questionnaire (FFQ) administered by a registered dietitian. In brief, nutrient intakes were based on the Chinese Food Composition Tables. The total dietary intake of each nutrient was calculated by summing the nutrients from all food items reported in the FFQs. Nutrient densities, the percentages of total energy from proteins, lipids, and carbohydrates, were calculated separately [30,31].

### 2.5. Statistical Analysis

Data were analyzed by using IBM SPSS Statistics 26 (IBM Corp., Armonk, New York, NY, USA). Descriptive statistics are shown as mean ± standard deviation for variables with a normal distribution and median (interquartile range [IQR]) for non-normally distributed variables, and descriptive statistics for nominal variables are shown as the number of cases (*n*) and percentage (%). Dietary data were all non-normally distributed. The differences between groups were investigated by analysis of variance (ANOVA) for normally distributed variables and the Kruskal–Wallis rank test for non-normally distributed variables. Nominal variables were assessed by Pearson’s chi-square or Fisher’s exact test. Variances in more than 2 groups were assessed by Tukey’s post hoc tests. Linear regression and logistic regression were used to assess the association of BMR and body composition parameters with sarcopenia and sarcopenic obesity. The area under the receiver operating characteristic curve was used to test the predictive accuracy of BMR for detecting sarcopenia or sarcopenic obesity and to determine an appropriate cutoff value [25,32].

Model 1 was an unadjusted model. Model 2 was adjusted for potential confounders (age and sex). ECW/ICW was added to Model 3 based on Model 2. ECW/TBW was added to Model 4 based on Model 2. Mediation and moderated mediation modeling analyses were performed with SPSS PROCESS v3.4 (Models 4 and 14, IBM Corp., Armonk, NY, USA). Optimal mediation effect testing was conducted by applying a bootstrapping procedure, which was used to measure indirect effects and corresponding 95% confidence intervals (CIs) with 5000 bootstrap samples. Age was also introduced into these models as a covariate. Results were considered statistically significant if *p* < 0.05 [11].

## 3. Results

### 3.1. Demographic, Clinical, and Body Composition Characteristics of Participants

A total of 402 geriatric patients were enrolled in this study. The male-to-female ratio was 240/162. According to the criteria, there were 20 (5.0%) subjects in the sarcopenia group. The prevalence of people in healthy, sarcopenic obesity, and obesity groups were 29.4% (118), 3.0% (12), and 62.7% (252), respectively.

The general characteristics of the four groups are displayed in Table 1 and Appendix A. The univariate analysis showed that participants in the sarcopenic obesity group were more likely to have a normal BMI, lower BMR, lower BMR/BMI, lower BMR/BSA, higher percent body fat, lower total skeletal muscle mass (TSM), lower ASM, and higher blood glucose. Similarly, the sarcopenia group had more participants older than 80 years (40.0%), who were more likely to have lower SMI, lower HGS, lower WHR, lower fat mass, lower BMR/Height^2^, higher ECW/ICW, higher ECW/TBW, and lower skeletal muscle mass. In contrast, participants with only obesity had higher VFA and the highest BMR, BMR/Height^2^, WHR, and fat mass.

### 3.2. Association between Dietary Components and Sarcopenia and Sarcopenic Obesity

The dietary intakes of total calories and main nutrients are shown in Table 2. The univariate analysis showed that participants in the obesity group had a higher carbohydrate density and a lower lipid density. Subjects with sarcopenia or sarcopenic obesity might be more inclined to intake fewer carbohydrates, especially those with sarcopenic obesity. No significant differences were found in total energy intake or in other macronutrient or micronutrient intakes.

The intake of macronutrients and micronutrients was separated into different groups according to Chinese Dietary Reference Intakes (DRIs) in Appendix A. Participants in the sarcopenic obesity group and obesity group showed a higher carbohydrate density in the >AMDR subgroup than in the normal group. There were no significant differences in other dietary components.

### 3.3. Association of Body Composition and Dietary Components with SMI

Two multivariate linear models were used to explore the relationship between body composition, dietary components, and SMI. BMR, body water distribution, total energy, and carbohydrate density were significantly correlated with SMI in both models. ECW/ICW and ECW/TBW were all negatively associated with muscle mass. BMR and total energy were positively correlated with SMI, while carbohydrate density showed a negative correlation with SMI. The details are shown below in Appendix A.

### 3.4. The Roles of BMR and Body Water Distribution in Predicting Sarcopenia and Sarcopenic Obesity

ROC curves were employed to explore the ability of BMR and body water to predict sarcopenia or sarcopenic obesity. The areas under the curve were all greater than 0.5.

The BMR cutoff points were 1271.2500 for diagnosing sarcopenic obesity and 1316.2570 for sarcopenia. BMR was also divided by BSA, BMI, and height squared (cm^2^) to exclude the effects of weight and height. BMR/BMI cutoff points were 56.8966 for sarcopenic obesity and 70.4416 for sarcopenia; BMR/BSA cutoff points were 810.778 for sarcopenic obesity and 832.2929 for sarcopenia; and BMR/height squared cutoff points were 0.0481 for sarcopenic obesity and 0.0479 for sarcopenia. ECW/ICW cutoff points were 0.6370 for sarcopenic obesity and 0.6348 for sarcopenia. ECW/TBW cutoff points were 0.3883 for sarcopenic obesity and 0.3891 for sarcopenia. Other ROC curve results are shown in Appendix A.

Cutoff values were used for logistic regression analysis, and the results are shown in Table 3. Subjects in the healthy group had higher BMRs (OR = 0.085, 95% CI 0.019–0.381, *p* = 0.001) compared to the sarcopenia group. After adjusting for age and gender, the effect remained significant (OR = 0.051, 95% CI 0.003–0.278, *p* = 0.001). The effect of BMR/BSA and BMR/Height^2^ in the sarcopenia group compared to the normal group was similar to the effect of BMR. Subjects in the healthy group also had higher BMRs (OR = 0.047, 95% CI 0.006–0.374, *p* = 0.004) compared to the sarcopenic obesity group. The effect was still significant after adjusting for age and gender. Participants in the healthy group also had higher BMR/BSA and BMR/BMI than the sarcopenic obesity group.

The roles of BMR/BMI, BMR/BSA, and BMR/Height^2^ in the relationship between sarcopenia and sarcopenic obesity were inconsistent. BMR/BMI and BMR/BSA were higher in the sarcopenia group. However, BMR/Height^2^ was higher in subjects with sarcopenic obesity.

After adding ECW/ICW to Model 3 and ECW/TBW to Model 4, the results changed somewhat (Appendix A). Lower BMR and higher ECW/ICW or ECW/TBW could increase the risk of sarcopenia (*p* < 0.05). After adjusting for body water, the protective effects of BMR and BMR/BMI on sarcopenic obesity were still significant. However, the predictive effects of other factors disappeared. The results also showed that ECW/ICW and ECW/TBW were higher in sarcopenic subjects than in sarcopenic obese people, while the effect of BMR disappeared after adjusting for body water.

### 3.5. Association of Carbohydrates with BMR and SMI

Appendix A presents the descriptive statistics and correlations of variables in this study. Carbohydrates were strongly correlated with BMR (r = 0.250, *p* < 0.001) and SMI (r = 0.222, *p* < 0.001) but did not correlate with body water distribution. SMI (r = −0.273, *p* < 0.001) and BMR (r = −0.244, *p* < 0.001) were both negatively correlated with body water distribution.

Figure 1 also shows an interesting phenomenon. Carbohydrate intake and BMR were positively correlated in the overall population (Figure 1a,b), and the same result was found for carbohydrate intake and SMI. The association in different groups was not consistent. Carbohydrate intake was also positively related to BMR and SMI in the healthy, sarcopenia, and obesity groups, but the opposite result was observed in the sarcopenic obesity group (Figure 1c,d). This might be associated with the extremely lower intake of energy and carbohydrates in the people in this group.

### 3.6. Mediation Analysis for Body Water Distribution, BMR, and SMI

Figure 2 shows that the positive relationship between carbohydrates and SMI was totally mediated by BMR when using the moderated mediation model (Model 4). The overall effect of carbohydrates on SMI was significant (β = 0.0033, *p* = 0.0003). The direct effect of carbohydrates on SMI was not significant (β = −0.0007, *p* = 0.0546) after controlling for age. The indirect effect of carbohydrates on SMI through BMR was significant, where indirect effect = 0.0039 (95% CI (0.0018, 0.0061)). This finding shows that BMR fully mediated the positive relationship between carbohydrates and SMI.

Moderated mediation analyses were conducted to further assess the relationship between body water distribution, carbohydrates, BMR, and SMI (Model 14). The specifications of these models can be found in Table 4. The results revealed a significant interactive effect of BMR and body water distribution on SMI. Figure 3 shows that the effect of carbohydrates on SMI through BMR varied with body water distribution. The index of moderated mediation was 0.0173 (Boot 95% CI 0.0068, 0.0321) for ECW/ICW and 0.0468 (Boot 95% CI 0.0185, 0.0858) for ECW/TBW (Appendix A). The effect of BMR on SMI was more obvious in the population with higher ECW/ICW or ECW/TBW. 

The final models are shown in Figure 4a,b. The moderated mediation analysis showed that BMR still fully mediated the relationship between carbohydrates and SMI after including the moderator. ECW/ICW (β = 0.0236, *p* < 0.001) or ECW/TBW (β = 0.0639, *p* < 0.001) moderated the second half of the path in the mediation models. Therefore, the positive relationship between carbohydrates and SMI was fully achieved through BMR, and these associations could be moderated by body water distribution.

## 4. Discussion

The co-occurrence of obesity and sarcopenia has become more and more common in adults aged 65 and older [6]. Patients with this condition, currently defined as sarcopenic obesity, are at risk for synergistic complications of sarcopenia and obesity. It is a high-risk geriatric syndrome that is mainly observed in the elderly population. A recent study found that sarcopenic obesity has a strong association with unfavorable outcomes and mortality in cancer patients [25]. Similar findings were observed in a 24-year follow-up cohort, which showed that sarcopenic obesity is clinically useful as a predictor of all-cause mortality [33]. Sarcopenic obesity is also related to cognitive impairment [29], low bone mineral density [26], frailty [7], and metabolic diseases [34]. A study found that carbohydrate intake could be the best indicator for determining sarcopenic obesity in older adults [35], and body composition plays an important role in sarcopenia. In this case, we explored the associations of nutrition and body composition with sarcopenia or sarcopenic obesity, evaluated the ability of BMR and body water distribution to identify sarcopenia or sarcopenic obesity, and found a significant mediating effect of BMR in the relationship between carbohydrates and SMI, as well as the moderating role of body water distribution in the indirect relationship between carbohydrates and SMI through BMR.

Participants in the sarcopenia and sarcopenic obesity groups were older than the rest of the population. This phenomenon has also been found in older Indian adults [26]. The results also showed that people with sarcopenic obesity had a higher fat mass and visceral fat area than people with sarcopenia but lower fat mass and visceral fat area than those with obesity. Sarcopenia and obesity are considered multifactorial syndromes with various overlapping causes and feedback mechanisms presumed to be strongly interconnected and mutually aggravating [36]. The redistribution of fat to the intra-abdominal area due to muscle atrophy enhances the secretion of some pro-inflammatory cytokines. Adipose tissue inflammation also leads to the fatty infiltration of skeletal muscle. This is a vicious cycle that promotes the development of sarcopenic obesity [9].

It was noticed that sarcopenic and sarcopenic obese subjects had lower BMR, SMI, and HGS but tended to have a normal BMI. After excluding the effects of height and weight, the difference in BMR was still significant. According to a study of Chinese elderly people, sarcopenic obese people seem to have a normal body size but have the highest risk of insulin resistance [37]. These results might indicate that the population with sarcopenic obesity cannot be identified by BMI, and BMR could reflect sarcopenia and sarcopenic obesity more sensitively. The results also showed that lower BMR was effective in evaluating sarcopenia. The suitable cutoff values of BMR, BMR/BSA, and BMR/Height^2^ might be simple screening tools for detecting sarcopenia. Lower BMR, BMR/BMI, and BMR/BSA might also detect sarcopenic obesity. These results are similar to a study in older males, which showed the roles of BMR and BMR/BSA in screening for sarcopenia [11]. After adding body water, the negative association of BMR with sarcopenia or sarcopenic obesity was still significant. However, the role of BMR in the differentiation of sarcopenia from sarcopenic obesity was unclear and needed to be further explored.

In addition, the study showed that body water might also affect sarcopenia and sarcopenic obesity, which has been confirmed by some research. Skeletal muscle tissue contains a large amount of water, which is partitioned into ICW and ECW fractions. Recent studies show that muscle quality, expressed as the ECW/ICW ratio, might reflect changes in muscle properties more sensitively than muscle quantity [14]. Low ICW and high ECW were correlated with muscle wasting and worse survival [38]. Higher ECW/TBW indicated overhydration, which might impair muscle strength [39]. ECW/TBW was also related to sarcopenia and might be a valid tool to evaluate muscle strength and physical performance in the elderly [40]. Our study also showed that ECW/ICW and ECW/TBW were both negatively correlated with SMI.

A study in older Korean adults showed an inverse relationship between carbohydrate intake and sarcopenic obesity [35]. Our study also found that the obese population took in more carbohydrates, but people with sarcopenic obesity had lower carbohydrate intake. The results showed that participants in the sarcopenic obesity group and obesity group had a higher carbohydrate density (>AMDR). Total energy intake was lower in the sarcopenic obesity group, which might have caused the low carbohydrate intake and high carbohydrate density. If the energy intake level was constant, the increase in carbohydrate intake would continue increasing the carbohydrate density, which might lead to a negative association of carbohydrate intake with BMR and SMI in the sarcopenic obesity group. In addition, carbohydrate intake significantly positively affected BMR and SMI in the sarcopenia group (Figure 1c,d). It seems that the overconsumption of carbohydrates (>AMDR) may influence fat distribution and might be a risk factor for obesity. These might indicate that different types of obesity have different mechanisms. Higher carbohydrate intake is related to an increase in BMR/kg in women with overweight or obesity [41]. This coincides with our study in that “obese people eat more carbohydrates that did not suffer from muscle loss, while people with sarcopenia and sarcopenic obesity intake less carbohydrates intake”. Therefore, we further evaluated the relationship between BMR and muscles and tried to find the roles of BMR and body water in the association of nutrition with muscle health.

Most of the population with sarcopenic obesity is sedentary. Small changes in their muscle mass can significantly alter daily energy expenditure and body water distribution and lead to lower BMR. In turn, these changes also exacerbate the vicious cycle in their metabolic development [6]. A recent study also showed that handgrip strength had a positive association with BMR [10]. Combined with the previous results, it could give support for the mediating role of BMR between carbohydrates and SMI. A long-term reduction in carbohydrate intake may cause the transfer of ICW to the extracellular space, resulting in high ECW/ICW and ECW/TBW [42]. This could support the moderating effects of body water distribution on the indirect relationship between carbohydrates and SMI through BMR. Glycogen depletion is often accompanied by TBW loss or ICW loss [43], which might result in high ECW/TBW or ECW/ICW. In this case, carbohydrate intake has a positive effect on glycogen synthesis [44], which might promote muscle performance. The effect of carbohydrates on the muscle through BMR also became more obvious at a higher level of body hydration in this study. Combined with the positive associations of carbohydrates with SMI and BMR in the sarcopenia group, the results showed that the benefit of proper carbohydrates for muscles was more effective in sarcopenia patients with higher ECW/ICW or ECW/TBW. Therefore, the role of muscle glycogen could be further studied, which could provide a new perspective on the relationship between nutrition and sarcopenia.

This study also had some limitations. First, due to the cross-sectional design, we could not draw conclusions about the etiological link between nutrients and sarcopenia or sarcopenic obesity, and the possible mechanism was not elucidated. In addition, the relationship might not be fully studied because of the small sample size of some groups, especially the sarcopenic obesity group. Finally, the sample only contained the Hainanese elderly population, so the results may not be applicable to other populations. Further studies are needed to clarify whether the results of the present study can be generalized to other populations and to elucidate the relevant mechanisms.

## 5. Conclusions

Higher BMR and lower ECW/ICW and ECW/TBW may benefit muscle health in elderly people. Lower BMR, especially lower BMR/BSA, has a potential role in predicting a higher risk of sarcopenia or sarcopenic obesity in the elderly. The dietary intake of carbohydrates had effects on muscle quality, muscle quantity, and fat distribution. The overconsumption of carbohydrates (especially > AMDR) might be a risk factor for obesity. However, moderate dietary carbohydrate intake could promote SMI by regulating BMR and body water distribution in the elderly. Due to the moderating role of ECW/ICW and ECW/TBW in the relationship of carbohydrates and BMR with SMI, the benefit of proper carbohydrates for muscles was more effective in sarcopenia patients with higher ECW/ICW or ECW/TBW.

## Figures and Tables

**Figure 1 nutrients-14-03911-f001:**
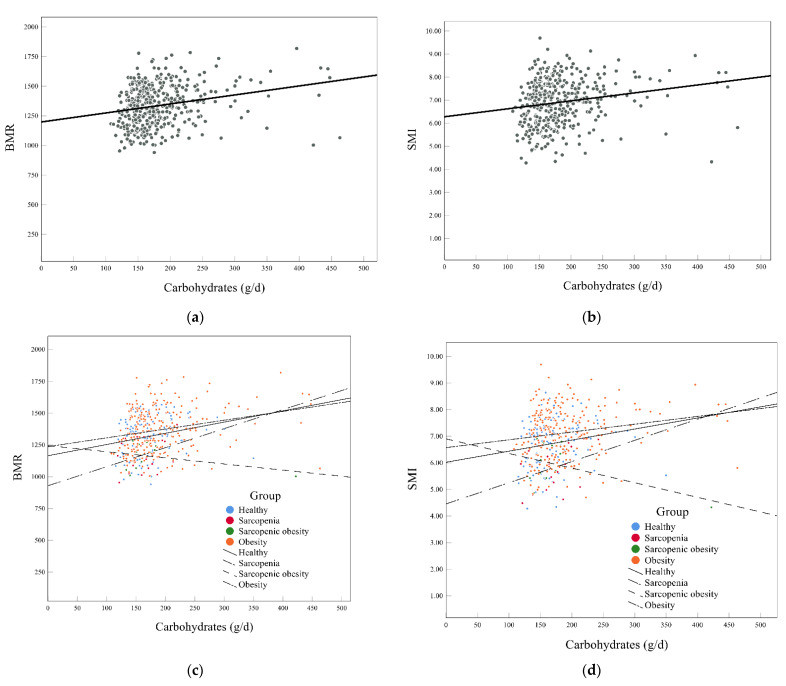
Associations of carbohydrates with BMR and SMI. (**a**,**c**) The association between carbohydrates and BMR in all subjects (**a**) and in different groups (**c**); (**b**,**d**) the association between carbohydrates and SMI in all subjects (**b**) and in different groups (**d**). BMR, basal metabolic rate; SMI, appendicular skeletal muscle index.

**Figure 2 nutrients-14-03911-f002:**
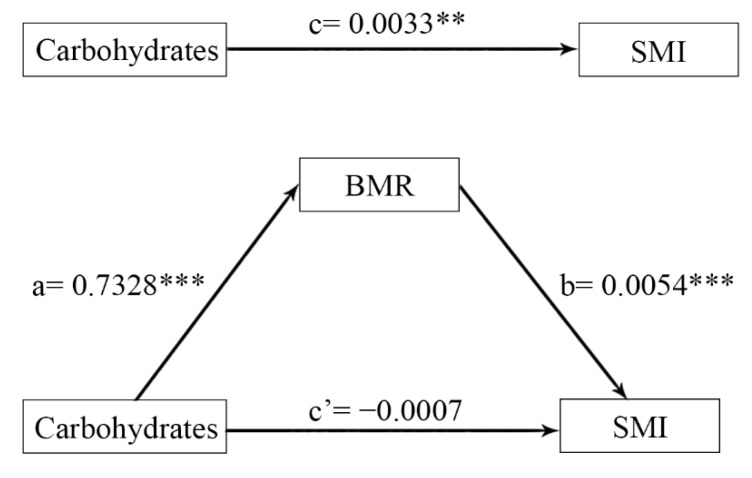
Mediating effect diagram for carbohydrates, BMR, and SMI while controlling for age. a×b: Indirect effect; c: total effect; c’: direct effect; c = c’ +  a×b. BMR, basal metabolic rate; SMI, appendicular skeletal muscle index. ** *p* < 0.01, *** *p* < 0.001.

**Figure 3 nutrients-14-03911-f003:**
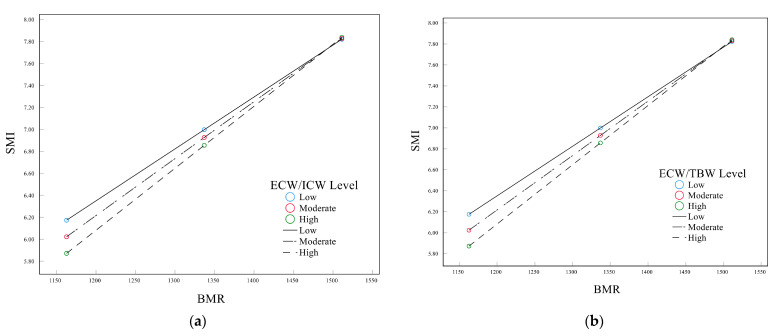
The moderating effects of BMR and body water distribution on SMI. (**a**) ECW/ICW; (**b**) ECW/TBW.

**Figure 4 nutrients-14-03911-f004:**
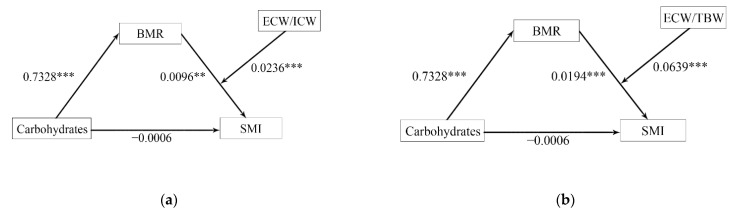
The moderated mediation models of BMR and body water distribution while controlling for age. (**a**) ECW/ICW; (**b**) ECW/TBW. a×b: Indirect effect; c: total effect; c’: direct effect; c = c’ +  a×b. BMR, basal metabolic rate; TBW, total body water; ICW, intracellular water; ECW, extracellular water; SMI, appendicular skeletal muscle index. ** *p* < 0.01, *** *p* < 0.001.

**Table 1 nutrients-14-03911-t001:** Body composition characteristics.

Characteristics	Total	Healthy	Sarcopenia	Sarcopenic Obesity	Obesity	*p*
BMR, (kcal/d)	1338.46 ± 174.40	1352.13 (1167.34, 1446.96) ^a,b,c^	1182.07 ± 133.69 ^a,d^	1158.01 ± 102.75 ^b,e^	1373.26 (1230.35, 1505.21) ^c,d,e^	<0.001 ***
BMI, (kg/m^2^)	24.07 ± 3.26	21.51 ± 2.22 ^a,b^	19.15 ± 2.08 ^a,c,d^	23.53 ± 2.91 ^c,e^	25.47 (24.00, 27.08) ^b,d,e^	<0.001 ***
BMR/BMI	56.17 ± 8.22	61.51 ± 7.54 ^a,b^	62.31 ± 8.87 ^c,d^	49.59 ± 5.14 ^a,c^	53.50 ± 6.98 ^b,d^	<0.001 ***
BMR/BSA	800.12 (762.31, 832.94)	827.77 ± 42.22 ^a,b^	821.76 (792.43, 829.97) ^c^	753.63 ± 42.38 ^a,c^	787.25 ± 45.84 ^b^	<0.001 ***
BMR/Height^2^	0.051 ± 0.003	0.050 ± 0.003 ^a,b^	0.047 ± 0.003 ^a,c^	0.049 ± 0.003 ^d^	0.052 ± 0.003 ^b,c,d^	<0.001 ***
TBW	33.17 (28.14, 37.55)	33.53 (27.27, 36.71) ^a,b^	27.72 ± 4.60 ^a,c^	26.87 ± 3.55 ^b,d^	34.28 (29.28, 38.72) ^c,d^	<0.001 ***
ICW	20.36 (17.31, 23.00)	20.54 (16.78, 22.73) ^a,b^	16.84 ± 2.82 ^a,c^	16.40 ± 2.22 ^b,d^	20.90 ± 3.70 ^c,d^	<0.001 ***
ECW	12.78 ± 2.28	12.88 (10.49, 14.16) ^a,b^	10.89 ± 1.80 ^a,c^	10.46 ± 1.36 ^b,d^	13.19 ± 2.28 ^c,d^	<0.001 ***
ECW/ICW	0.632 ± 0.001	0.629 ± 0.001 ^a^	0.647 ± 0.004 ^a,b^	0.639 ± 0.006	0.632 ± 0.001 ^b^	0.001 ***
ECW/TBW	0.387 ± 0.007	0.386 ± 0.006 ^a^	0.393 ± 0.006 ^a,b^	0.390 ± 0.007	0.387 ± 0.008 ^b^	0.001 ***
SMI, (kg/m^2^)						
Men	7.49 ± 0.69	7.28 ± 0.52 ^a,b,c^	6.31 ± 0.52 ^a,d^	6.59 ± 0.31 ^b,e^	7.74 ± 0.62 ^c,d,e^	<0.001 ***
Women	6.05 ± 0.74	5.64 ± 0.57 ^a,b^	5.03 ± 0.41 ^a,c^	5.23 ± 0.53 ^d^	6.31 ± 0.65 ^b,c,d^	<0.001 ***
HGS, (kg)						
Men	34.20 ± 7.33	35.17 ± 5.84 ^a,b^	22.60 ± 3.36 ^a,c^	24.82 ± 3.37 ^b,d^	35.03 ± 7.34 ^c,d^	<0.001 ***
Women	23.00 (20.15, 24.80)	23.00 ± 2.68 ^a,b^	15.16 ± 2.33 ^a,c^	14.87 ± 1.31 ^b,d^	23.25 (20.83, 24.88) ^c,d^	<0.001 ***
Waist, (cm)	88.00 (80.50, 93.00)	78.00 (75.00, 85.00) ^a^	75.00 ± 5.69 ^b^	82.83 ± 5.61 ^c^	92.00 (88.00, 96.00) ^a,b,c^	<0.001 ***
Hip, (cm)	93.63 ± 5.34	89.93 ± 3.87 ^a,b^	85.76 ± 3.23 ^a,c,d^	89.89 ± 2.65 ^c,e^	96.18 ± 4.34 ^b,d,e^	<0.001 ***
WHR						
Men	0.896 ± 0.066	0.854 ± 0.045 ^a^	0.836 ± 0.055 ^b^	0.855 ± 0.052 ^c^	0.925 ± 0.061 ^a,b,c^	<0.001 ***
Women	0.879 (0.858, 0.18)	0.860 (0.829, 0.871) ^a^	0.828 ± 0.035 ^b,c^	0.899 ± 0.047 ^b^	0.900 ± 0.044 ^a^^,c^	<0.001 ***
FM, (kg)	17.74 ± 5.95	12.46 ± 3.81 ^a,b^	10.81 ± 3.01 ^c,d^	17.14 ± 3.32 ^a,^^c^^,e^	20.81 ± 4.72 ^b,d^^,e^	<0.001 ***
Percentage Body Fat, (%)	28.41 (23.18, 33.43)	22.31 (18.64, 25.74) ^a,b^	22.39 ± 6.07 ^c,d^	32.07 ± 6.03 ^a,c^	31.06 ± 5.95 ^b,d^	<0.001 ***
VFA, (cm^2^)	81.78 (64.15, 107.55)	57.60 ± 19.41 ^a,b^	53.36 ± 15.43 ^c,d^	88.29 ± 26.98 ^a,c^	97.49 (79.77, 118.49) ^b,d^	<0.001 ***
TSM, (kg)	24.59 (20.58, 28.01)	24.79 (19.89, 27.64) ^a,b^	19.96 ± 3.68 ^a,c^	19.39 ± 2.89 ^b,d^	25.30 ± 4.83 ^c,d^	<0.001 ***
ASM, (kg)	18.45 (15.01, 21.23)	18.68 (14.57, 21.02) ^a,b^	14.83 ± 3.33 ^a,c^	14.00 ± 2.41 ^b,d^	18.85 ± 3.98 ^c,d^	<0.001 ***

Continuous variables are shown as mean ± standard deviation for variables with normal distributions and median (interquartile range [IQR]) for non-normal distributions. Categorical variables are shown as *n* (%). BMI, body mass index; BMR, basal metabolic rate; SMI, appendicular skeletal muscle index; HGS, handgrip strength; WHR, waist–hip ratio; FM, fat mass; VFA, visceral fat area; TSM, total skeletal muscle mass; ASM, appendicular skeletal muscle; BSA, body surface area; TBW, total body water; ICW, intracellular water; ECW, extracellular water. ^a,b,c,d,e^ Indicates *p* < 0.05 between two means or distributions with the same uppercase letter in the same line. *** *p* < 0.001.

**Table 2 nutrients-14-03911-t002:** Dietary intakes of different populations.

	Total	Healthy	Sarcopenia	Sarcopenic Obesity	Obesity	*p*
Total energy (kcal/d)	1442.50 (1228.07, 1715.92)	1401.03 (1215.04, 1651.26)	1523.75 (1189.21, 1644.23)	1285.54 (1169.97, 1652.64)	1470.18 (1238.49, 1756.24)	0.353
Macronutrients						
Carbohydrates (g/d)	168.86 (148.71, 201.16)	164.50 (144.71, 187.70) ^a^	165.39 (139.66, 193.49)	157.20 (145.97, 179.80)	173.20 (150.43, 209.10) ^a^	0.023 *
Carbohydrate density (%E)	48.11 (42.95, 53.51)	46.69 (41.81, 51.82)	46.18 (43.68, 51.38)	48.83 (42.49, 54.90)	49.36 (43.56, 54.19)	0.101
Total proteins (g/d)	53.52 (44.44, 65.30)	54.62 (44.06, 63.56)	57.84 (44.26, 69.87)	48.07 (40.81, 56.13)	53.21 (44.86, 65.70)	0.475
Total protein density (%E)	15.18 (13.69, 16.72)	15.20 (13.84, 16.85)	15.75 (13.64, 19.78)	14.17 (12.65, 17.05)	15.15 (17.66, 16.43)	0.397
Lipids (g/d)	79.70 (65.86, 96.25)	81.62 (67.65, 95.88)	81.15 (67.93, 89.02)	71.58 (67.25, 99.85)	78.26 (65.30, 96.83)	0.895
Lipid density (%E)	50.23 (45.77, 54.38)	52.09 (47.45, 55.15) ^a^	50.02 (46.63, 54.06)	51.67 (48.14, 54.40)	49.62 (44.24, 53.66) ^a^	0.028 *
Dietary fiber (g/d)	13.63 (11.38, 16.46)	12.80 (11.24, 15.67)	13.99 (11.63, 16.48)	13.95 (11.62, 15.51)	13.69 (11.41, 16.86)	0.553
Micronutrients						
Potassium (mg/d)	1811.75 (1534.11, 2172.01)	1743.51 (1494.08, 2049.12)	2131.51 (1536.93, 2496.12)	1593.52 (1517.71, 2335.03)	1843.12 (1555.04, 2207.02)	0.143
Magnesium (mg/d)	307.96 (266.48, 373.41)	295.39 (261.77, 353.80)	347.26 (284.25, 391.39)	303.52 (249.45, 370.94)	309.36 (268.15, 380.47)	0.411
Manganese (mg/d)	5.68 (5.29, 6.22)	5.58 (5.20, 6.20)	5.48 (5.21, 5.83)	5.61 (5.07, 6.13)	5.75 (5.37, 6.46)	0.060
Phosphorus (mg/d)	920.98 (769.18, 7090.43)	895.61 (747.90, 1061.79)	969.23 (772.61, 1129.42)	786.57 (731.82, 998.35)	925.40 (782.35, 1131.52)	0.249
Ferrum (mg/d)	18.94 (17.13, 22.47)	18.32 (16.56, 21.92)	19.45 (17.55, 22.15)	18.75 (16.10, 22.17)	19.18 (17.27, 22.89)	0.281
Calcium (mg/d)	575.02 (412.00, 751.03)	533.23 (364.90, 708.55)	643.82 (476.83, 756.19)	611.03 (353.74, 775.06)	580.38 (425.21, 773.83)	0.240
Vitamin A (µg/d)	569.50 (267.51, 1078.39)	712.40 (272.77, 1107.14)	769.44 (326.50, 1020.27)	459.68 (246.40, 1022.73)	476.91 (256.80, 1086.06)	0.855
Vitamin B1 (µg/d)	0.74 (0.64, 0.89)	0.74 (0.61, 0.89)	0.74 (0.65, 0.85)	0.66 (0.62, 0.87)	0.75 (0.64, 0.90)	0.447
Vitamin B2 (µg/d)	3.49 (3.34, 3.70)	3.48 (3.35, 3.73)	3.49 (3.33, 3.70)	3.32 (3.29, 3.52)	3.50 (3.35, 3.70)	0.172
Vitamin B3 (µg/d)	11.54 (9.76, 14.48)	11.60 (9.76, 13.99)	12.22 (9.97, 13.51)	10.60 (8.84, 13.67)	11.52 (9.74, 15.01)	0.721
Vitamin C (mg/d)	150.70 (81.67, 231.61)	146.50 (80.76, 225.09)	156.37 (100.32, 215.49)	190.33 (85.70, 212.57)	148.43 (81.20, 244.77)	0.948

Continuous variables are shown as median (interquartile range [IQR]) for non-normal distributions. ^a^ Indicates *p* < 0.05 between two means with the same uppercase letter in the same line. * *p* < 0.05.

**Table 3 nutrients-14-03911-t003:** Associations of BMR with sarcopenia and sarcopenic obesity.

	Healthy vs. Sarcopenia		Healthy vs. Sarcopenic Obesity		Sarcopenia vs. Sarcopenic Obesity	
	OR (95% CI)	*p*	OR (95% CI)	*p*	OR (95% CI)	*p*
BMR						
MODEL 1	0.085 (0.019, 0.381)	0.001 **	0.047 (0.006, 0.374)	0.004 **	0.244 (0.042, 1.414)	0.116
MODEL 2	0.051 (0.009, 0.278)	0.001 **	0.007 (0.000, 0.087)	<0.001 ***	0.195 (0.022, 1.726)	0.142
BMR/BMI						
MODEL 1	2.476 (0.780, 7.865)	0.124	0.031 (0.004, 0.250)	0.001 **	0.030 (0.003, 0.297)	0.003 **
MODEL 2	3.262 (0.800, 13.310)	0.099	0.003 (0.000, 0.051)	<0.001 ***	—	0.999
BMR/BSA						
MODEL 1	0.100 (0.022, 0.452)	0.003 **	0.035 (0.004, 0.284)	0.002 *	0.083 (0.015, 0.459)	0.004 **
MODEL 2	0.060 (0.011, 0.319)	0.001 **	—	0.998	—	0.999
BMR/Height^2^						
MODEL 1	0.135 (0.045, 0.400)	<0.001 ***	0.439 (0.133, 1.454)	0.178	7.500 (1.288, 43.687)	0.025 *
MODEL 2	0.152 (0.045, 0.510)	0.002 **	0.668 (0.151, 2.962)	0.596	22.507 (1.792, 282.647)	0.016 *

MODEL 1: Unadjusted model; MODEL 2: adjusted for potential confounders (age and sex). BMI, body mass index; BMR, basal metabolic rate; BSA, body surface area. * *p* < 0.05, ** *p* < 0.01, *** *p* < 0.001.

**Table 4 nutrients-14-03911-t004:** Moderated mediation effect analysis.

Predictors	On BMR			On SMI		
	Coeff	t	*p*	Coeff	t	*p*
ECW/ICW						
Control variables						
Age	−4.2375	−3.3246	0.0010 **	0.0004	0.1474	0.8829
Independent variable						
Carbohydrates	0.7328	4.7466	<0.0001 ***	−0.0006	−1.7520	0.0806
Mediator						
BMR				−0.0096	−3.3917	0.0008 ***
Moderator						
ECW/ICW				−35.3383	−5.5645	<0.0001 ***
Interaction term						
BMR × ECW/ICW				0.0236	5.2782	<0.0001 ***
ECW/TBW						
Control variables						
Age	−4.2345	−3.3246	0.0010 **	0.0004	0.1252	0.9004
Independent variable						
Carbohydrates	0.7328	3.0519	<0.0001 ***	−0.0006	−1.7511	0.0807
Mediator						
BMR				−0.0194	−4.1371	<0.0001 ***
Moderator						
ECW/TBW				−95.4461	−5.5599	<0.0001 ***
Interaction term						
BMR × ECW/TBW				0.0639	5.2767	<0.0001 ***

Coeff, the standardized regression coefficient; LLCI, lower-limit confidence interval; ULCI, upper-limit confidence interval; BMR, basal metabolic rate; TBW, total body water; ICW, intracellular water; ECW, extracellular water; SMI, appendicular skeletal muscle index. ** *p* < 0.01, *** *p* < 0.001.

## Data Availability

Data are available upon reasonable requests.

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
