# Peer review of "Predictive Roles of Basal Metabolic Rate and Body Water Distribution in Sarcopenia and Sarcopenic Obesity: The link to Carbohydrates"

_nutrients, 2022, doi:10.3390/nu14193911_

Round 1
Reviewer 1 Report
nutrients-1890732
Title: Predicting Roles of Basal Metabolic Rate and Body Water Distribution in Sarcopenia and Sarcopenic Obesity: The link of Carbohydrates
Authors: L Guan, T Li , X Wang, K Yu, R Xiao, Y Xi
These researchers evaluated the relationship of basal metabolic rate (BMR) and body water distribution in 402 elderly subjects on muscle health and their prospective roles on screening sarcopenic obesity and sarcopenia. Body composition was estimated by bioelectrical impedance analysis. The intake of dietary carbohydrates in people with sarcopenic obesity was the lowest, but in subjects with obesity was the highest. The results of moderated mediation model showed BMR fully mediated the positive relationship between carbohydrates and SMI, which would be more obvious in the population with abnormal body water distribution. BMR or BMR/body surface area (BSA) had the potential role of predicting higher risk of sarcopenic obesity and sarcopenia. Higher BMR, lower extracellular water (ECW)/intracellular water (ICW) and ECW/total body water (TBW) may benefit muscle health. Lower dietary carbohydrates intake had no beneficial effects on muscle health. Moderately high dietary carbohydrates intake might promote skeletal muscle index (SMI) through regulating BMR and body water distribution in elderly. The authors demonstrated that higher BMR, lower ECW/ICW and ECW/TBW may benefit muscle health in elderly people, whereas lower BMR, especially lower BMR/BSA had the potential role of predicting higher risk of sarcopenic obesity or sarcopenia in the elderly. Furthermore, the dietary intake of carbohydrates affects muscle health and fat distribution.
However, lower carbohydrates intake had no beneficial effects on muscle health, which might be contribute to sarcopenic obesity. Moreover, overconsumption of carbohydrates (>AMDR) might be the risk factor of simple obesity, and moderately high dietary carbohydrates intake might promote SMI through regulating BMR and body water distribution in elderly population. Also, the much higher of ECW/ICW and ECW/TBW, the greater effects of carbohydrates and BMR on SMI could be found in present study.
The researchers made a confusing statement by commenting that dietary intake of carbohydrates affects muscle health and fat distribution, while lower carbohydrates intake had no beneficial effects on muscle health, which might be contribute to sarcopenic obesity.
Reviewer 2 Report
I am glad for the opportunity to review the article entitled "Predicting roles of basal metabolic rate and body water distribution in sarcopenia and sarcopenic obesity: the link of carbohydrates".
The authors are demonstrating in this manuscript the relationship of basal metabolic rate, body water distribution and carbohydrates intakes on elderly people, which is an interesting link. However, I have few points to highlight to improve the article.
1) Introduction is well-written and is giving a good background to the readers.
2) Methods is also well-written, and it is clear which parameters and tools are used by the authors to achieve the objective.
3)Results:
- Table 1: Please, add the BMI data and also the word obesity in the sarcopenic column.
-In the final models (figure 2), add a paragraph explain it.
4) The authors did a good job in the discussion section.
Overall, the article has a good subject, however it needs to improve the results presentation. I believe that adding graphs to demonstrate some data is more clear, ie try to use a linear regression graph to show the BMR and carbohydrate intake, or BMR and SMI, and so on.
Round 2
Reviewer 1 Report
The Manuscript reads well and now recommended for acceptance.